# The TCGA Molecular Classification of Endometrial Cancer and Its Possible Impact on Adjuvant Treatment Decisions

**DOI:** 10.3390/cancers13061478

**Published:** 2021-03-23

**Authors:** Matthias Alexa, Annette Hasenburg, Marco Johannes Battista

**Affiliations:** Department of Gynecology and Obstetrics, University Medical Center Mainz, 55131 Mainz, Germany; annette.hasenburg@unimedizin-mainz.de (A.H.); battist@uni-mainz.de (M.J.B.)

**Keywords:** uterine cancer, uterine carcinoma, endometrial cancer, endometrial carcinoma, radiotherapy, radiobiology, brachytherapy

## Abstract

**Simple Summary:**

The Proactive Molecular Risk Classifier for Endometrial Cancer (ProMisE) is a recently developed tool to identify four distinct molecular subgroups of endometrial cancer. Patients identified as Polymerase Epsilon exonuclease domain mutated (POLE EDM) or p53-mutated have significantly altered prognosis compared to patients allocated to the mismatch repair deficient (MMRd) or p53 wt groups. The aim of this review is to give a broad overview over the initial development and refinement of the classifier as well as possible effects on the recommended adjuvant treatment. We have summarized the clinical data of 8 studies including 3650 endometrial cancer patients and analyzed the distribution of tumor stage and adjuvant treatment received in respect to the molecular subgroups. Based on the findings of the summarized studies treatment de-escalation might be feasible for POLE EDM patients while p53 abn patients should receive adjuvant (chemo-)radiotherapy.

**Abstract:**

Adjuvant treatment decisions for endometrial cancer (EC) are based on stage, the histological grade of differentiation, histological subtype, and few histopathological markers. The Proactive Molecular Risk Classifier for Endometrial Cancer (ProMisE) identified four risk groups of EC patients using a combination of immunohistochemistry and mutation analysis: Polymerase Epsilon exonuclease domain mutated (POLE EDM), mismatch repair deficient (MMRd), p53 wild-type/copy-number-low (p53 wt), and p53-mutated/copy-number-high (p53 abn). Patients allocated to the POLE or abnormal p53 expression subtype are faced with a significantly altered outcome possibly requiring a modified adjuvant treatment decision. Within this review, we summarize the development of ProMisE, characterize the four molecular subtypes, and finally discuss its value in terms of a patient-tailored therapy in order to prevent significant under or overtreatment.

## 1. Introduction

Endometrial cancer (EC) is the 6th most common cancer in females. Roughly 382,000 new cases are diagnosed each year and EC accounts for 90,000 deaths worldwide [1]. The estimated five-year survival rate in developed countries is about 80% with primary treatment consisting of combinations of surgery, vaginal brachytherapy (VBT), external beam radiation therapy (EBRT), and adjuvant chemotherapy [2,3]. Treatment decisions are based on the tumor stage and pathological findings. Historically two types of EC have been described by Bokhman in 1983 [4]. Bokhman specified a more common, favorable type associated with obesity, metabolic syndrome, and hyperestrogenism. The second type developing independently of risk factors known at that time. More recently, it became clear that this distinction, while easily applicable, is not ideal for the heterogeneous group of EC, especially high-grade EC [5,6]. To compensate for these shortcomings the world health organization (WHO) classifies EC determining the expression of various markers into four histological types: low-grade endometrioid, high-grade endometrioid, serous, and clear cell EC [7]. However, the histopathological findings are prone to a misdiagnosis as multiple studies have shown uncertainties between pathologists in about 30% of the cases [8,9]. Various methods of risk group classifications are in place and recommended by international societies to determine patients that need adjuvant treatment [3,10,11]. This review touches on the conventional histo-pathological classifier, several selected single prognostic markers, and elucidates the development of the Proactive Molecular Risk Classifier for Endometrial Cancer (ProMisE) and its possible implications on adjuvant treatment decisions.

## 2. Development of a New Molecular Classification

A vast number of molecular and histological markers or mutations have been studied to determine their relevance to the risk and outcome of early endometrial cancer.

The L1 cell adhesion molecule (L1CAM) is a 200–220 kD transmembrane protein that is overexpressed in 7–18% of EC [12,13]. Retrospective studies have shown a significant correlation to distant recurrence and overall survival for tumors with >10% L1cam expression [14,15].

Lymphovascular space invasion (LVSI) has been identified as an independent predictor for local as well as distant recurrence [16,17,18]. LVSI occurred more frequently in high-grade tumors and tumors with deep (>50%) myometrial invasion. LVSI has since been established as a decisive predictor to determine the need for adjuvant therapy in early, low-grade (1–2) endometrial cancer [3,10].

The β-Catenin gene (CTNNB1) has been detected as a frequent mutation in EC and a mutational hot spot in multiple cancer types [19]. The CTNNB1 mutation has been associated with increased recurrent disease in low-grade, early EC and a worse overall survival in some studies, while others did not show a significant difference [20,21,22].

Prolonged exposure to estrogen was identified as a risk factor for developing endometrial hyperplasia and endometrial cancer a long time ago [23]. Historically the estrogen dependence was suggested as a distinction between favorable estrogen-dependent and unfavorable estrogen-independent endometrial cancers [4,24]. Loss or under-expression of estrogen receptors is associated with a worse overall outcome [25]. While estrogen receptor is a major factor in breast cancer it has not been an important marker for EC [10].

The phosphatase and tensin homolog deleted on chromosome 10 (PTEN) is part of the PI3K/AKT/mTOR pathway regulation [26]. Loss or alteration of PTEN occurs in 45% of EC and is more commonly found in endometrioid EC than in other histological subtypes [27]. PTEN alterations are also detected in curettage specimens but no significant prognostic relevance for the progression to EC has been identified [28].

Considering the high number of possible markers, only a few have been included in internationally recommended guidelines for risk stratification (Table 1). The Cancer Genome Atlas (TCGA) performed a genome-wide analysis of 373 EC instead of focusing on a combination of singular well-studied molecular or histopathological markers [29]. In addition to exome sequencing, mRNA expression, protein expression, miRNA expression, and DNA methylation were evaluated. Four distinct groups have been identified: Polymerase Epsilon ultra-mutated, microsatellite instability hypermutated, copy-number-low, and, copy number high EC. Derived from this data Talhouk et al. [30,31] developed ProMisE to identify similar subgroups using a combination of immunohistochemistry (IHC) and mutation analysis instead of genomic data (Figure 1). In addition to the original mutation analysis, their work incorporated survival analysis, anticipated clinical benefit as well as cost and accessibility of methods into the algorithm. The ProMisE was extensively tested against new cohorts of EC patients over the following years, evaluating its power for determining risk groups and predicting the prognosis (Table 2) [25,30,31,32,33,34].

## 3. Main Features of the Molecular Subtypes

### 3.1. Polymerase Epsilon (POLE)

Polymerase Epsilon (POLE) is a DNA replicate with a proofreading domain responsible for the usually low mutation rate in eukaryotic DNA replication [36,37]. Mutations in the proofreading domain are found in several types of tumors and lead to a high yield of mutational frequency and low somatic copy-number alterations (SCNA) [29,35]. Multiple pathogenic and non-pathogenic mutations have been reported for EC including a recommendation for interpreting non-hotspot mutations [38]. The final decision which mutations define polymerases epsilon exonuclease domain mutated (POLE EDM) tumors is a topic of ongoing research and has yet to be answered. This group was defined as ultra-mutated or polymerases epsilon exonuclease domain mutated and includes 4–12% of EC [25,32,33,34]. POLE EDM tumors were more commonly found in younger women with a lower body mass index (BMI). Although tumors were predominantly assessed as high-grade, the prognosis for these patients was excellent. These tumors mostly present in earlier stages with an early onset of symptoms.

### 3.2. Mismatch Repair Deficiency (MMRd)

Acquired mismatch repair deficiency resulting in microsatellite instability (MSI) is a diagnostic phenotype for a multitude of cancer types [39]. MSI is of clinical relevance, especially for colorectal cancer. Either germline mutations in the key genes MLH1, MSH2, MSH6, or PMS2 as described for the Lynch syndrome or somatic mutations can result in MMRd tumors [40,41,42]. The group of MMRd accounts for 23–36% of EC while only 2% of all EC are associated with Lynch syndrome. The mutation frequency is considerably high but not as high as the POLE EDM group. Frequent gene mutations in this group are PTEN, PIK3CA, and PIK3R1 [35]. Substantial LVSI is more often present in MSI tumors.

### 3.3. p53 Abn

The tumor suppressor gene *TP53* which encodes for the p53 protein has been established as one of the most common mutations in human tumors [43,44]. This subgroup was initially defined by the high number of somatic copy-number alterations and low mutational yield but p53 IHC was later chosen as a representative method to determine tumors of this group [29,30]. In endometrial cancer p53 alterations to either overexpression or missense are associated with the worst prognosis of all molecular subtypes [45]. The group of p53 abn includes most of the serous and mixed types, presents at higher stages, and is more commonly grade 3. Nevertheless, lower grades and early stages are also found for p53 abn tumors. The p53 subgroup includes 8–24% of EC.

### 3.4. p53 wt

The group of EC that did not exhibit any of the previously described features was classified as p53 wt or no specific molecular profile (NSMP). The p53 wt subgroup was found to be rather heterogeneous since it covers the remaining tumors. Patients with p53 wt tumors exhibited a higher BMI (mean 33.7) [32]. With only a few exceptions, these tumors express estrogen and progesterone receptors and predominantly present as an endometrioid subtype. The majority of EC (30–60%) exhibits this subtype.

## 4. Published Trials

To date, the proposed classification has been applied to a total of 3650 patients diagnosed with EC in 8 studies (Table 3). These include the published cohorts for the discovery, confirmation, and validation of the new molecular classifier [30,31,46]. Other groups have evaluated the ProMisE against a great number of EC patients [25,33,34,47,48].

After an initial suggestion of the TCGA classification, Talhouk et al. aimed to develop an easier, molecular-based, and financially feasible method to identify the TCGA subgroups [30]. A retrospective cohort of 143 EC patients including all grades and stages was identified and extensively tested. The methods and algorithm that reproduced the subgroups and their clinical outcome in the best way were identified and is shown in Figure 1. An attempt to include a *PTEN* mutation into the algorithm did not yield any additional benefit. This cohort was considered the discovery cohort. Of the included 143 patients, 12 (9%) were grouped into POLE EDM, 41 (29%) into MMRd, 63 (45%) into p53 wt, and 25 (18%) were p53 abn. Adjuvant radiation was received by 63 (44%) of the patients including 5 patients and 24 patients of the POLE EDM group and MMRd group respectively. During the observation time (median 4.67 years) a total of 27 recurrences and 28 deaths occurred. POLE EDM and MMRd showed a better recurrence-free survival (RFS) (hazard ratio (HR): 0.1; 95% CI = 0.00–0.77 and HR: 0.5; 95% CI = 0.18–1.29, respectively) compared to the p53 wt group while p53 abn exhibited a worse RFS (HR 1.1; 95% CI = 0.45–2.64).

As a second step, the developed algorithm was evaluated in comparison with a larger confirmation cohort. Talhouk et al. [31] retrospectively analyzed 319 EC cases in this study. Patients were included regardless of stage, grading, and histology. In this cohort 30 (9%) patients were considered as POLE EDM, 64 (20%) as MMRd, 139 (44%) as p53 wt, and 86 (27%) were p53 abn. Close to half of the patients (47.4%) received adjuvant treatment including 18 (60%) and 28 (44%) of the POLE EDM and MMRd group, respectively. The ProMisE classifier showed comparable predictive value as the European Society for Medical Oncology (ESMO) risk classifier but when combined their discriminative ability was even better. In Talhouk’s study, 16 (3.4%) patients had more than one molecular feature but were categorized to a subtype by the algorithm shown in Figure 1. Progression-free survival (PFS) was worse in p53 abn (HR: 1.75; 95% CI = 0.84–3.96) patients compared to p53 wt. MMRd patients showed a slightly better PFS (HR: 0.64; 95% CI = 0.25–1.60) while the POLE EDM group had the best outcome (HR: 0.19; 95% CI = 0.02–0.81) of the molecular subgroups.

Lastly, a third validation cohort from a different tertial referral center was established [46]. A total of 452 consecutive cases were retrospectively included and analyzed. For 152 patients both hysterectomy and biopsy or curettage samples were available. The determined molecular profile in the diagnostic sample was highly consistent with the post-surgical specimen. Distribution among the subgroups was comparable to the previous studies. More than one molecular feature was detected in eight (1.8%) of the patients.

Additionally, Kommos et al. [49] identified L1CAM as an independent risk factor for the p53 wt group. The subgroup of p53 wt, L1CAM+ tumors, while associated with high tumor grade and high International Federation of Gynecology and Obstetrics (FIGO) stage, had a comparably poor outcome as p53 abn tumors. The effect of L1CAM positivity was insignificant for the other molecular groups.

Stello et al. [25] employed the algorithm on an existing cohort from the Postoperative Radiation Therapy in Endometrial Carcinoma (PORTEC)-1 and -2 trials. Of the total trial population (*n* = 1141) a successful analysis was possible in 861 patients. Exclusions were due to non-endometrioid endometrial cancer, missing materials, or failed analysis. A total of 27 (3.1%) samples showed multiple molecular features. Overall survival was significantly worse in p53 abn (HR: 3.777; 95% CI = 2.364–6.037) and MMRd (HR: 1.879; 95% CI = 1.307–2.700). The difference between POLE EDM (HR: 0.907; 95% CI: 0.367–2.237) and p53 wt (reference) was not as distinguished as in other studies. The authors also evaluated the relevance of other molecular and histopathological features. Significant impact on recurrence or survival was found only for substantial LVSI, >10% L1CAM expression or <10% ER and PR expression.

Bosse et al. established a cohort of 381 grade 3 endometrioid EC including all stages [34]. Notably, POLE EDM patients were significantly younger and the majority of this cohort (47/48) presented with a stage I tumor. Thus, additional multivariable analysis for RFS was performed for stage I only. The POLE EDM group had significantly better RFS (HR: 0.25; 95% CI = 0.07–0.83) compared to p53 wt. MMRd tumors showed a trend towards better RFS (HR: 0.59; 95% CI: 0.30–1.17) while p53 abn had a significantly worse outcome (HR: 2.44; 95% CI: 1.34–4.465).

Cosgrove et al. [33] evaluated 1040 EC from the NRG Oncology/Gynecologic oncology group (GOG) 210 trial. The outcome analysis showed similar results as the other published studies. Interestingly, during the follow-up time (median = 60.62 months) out of 84 p53 abn cases, 31 had recurrences, and 16 cancer-associated deaths occurred. In contrast among the 39 POLE EDM patients, only 2 recurrences and 1 cancer-associated death were reported. In the POLE EDM group, nine (23.7%) women received any adjuvant treatment. This is the highest rate of adjuvant treatment among the four groups.

Raffone et al. [50] conducted a meta-analysis of six studies including clinical data of 2818 EC patients. Of patients with available data, 75.4% (1711) were Stage I, and 24.6% (557) were Stage II–IV. Any adjuvant treatment was received by 47.2% (1283) patients. The multivariable analysis regarding PFS showed a risk 1.8 times higher for p53 abn compared to p53 wt. POLE EDM had a risk 5 times lower while MSI showed no significant difference. The results for overall survival (OS) showed a 2 times lower risk of death of any cause for POLE EDM compared to p53 wt. MSI and p53 abn showed a 1.5- and 3-times higher risk respectively. However, the multivariable analysis showed a 2 times higher risk for p53 abn, while no significant difference was detected for MSI and POLE EDM. The authors discussed the loss of significance at the multivariable analysis for the MSI group regarding OS, disease specific survival (DSS), and PFS.

Most recently León-Castillo et al. have published data of the PORTEC-3 trial [48]. For 423 of 660 patients, tissue samples were available. In this high-risk cohort, the distribution and outcome analysis for the molecular subgroups were consistent with other publications. The patients were equally distributed into the experimental (combined chemoradiotherapy) and control (EBRT) arms for the whole study population as well as for the molecular subgroups. Surprisingly MMRd was chosen as a reference for the multivariable outcome analysis thus making the comparison to other studies difficult. Newly the impact of adjuvant treatment was evaluated. The p53 abn subgroup had a significant (HR: 0.52; 95% CI = 0.30–0.91) benefit if treated with combined adjuvant chemotherapy and radiotherapy (5-year RFS: 58.6%) vs. radiotherapy alone (5-year RFS: 36.2%). All other subgroups did not show a significant difference. Of the POLE EDM group, only one patient had a recurrence resulting in a 5-year RFS of 96.6%.

A small number (3%) of EC exhibited features of more than one molecular profile [51]. A detailed molecular and genomic analysis was performed for the three groups MMRd-p53 abn, POLE EDM-p53 abn, and MMRd-POLE EDM-p53 abn. Upon review MMRd-p53 abn was revealed to behave more similarly to MMRd than p53 abn in terms of histopathological features and clinical outcome. Similarly, POLE EDM-p53 abn tumors were classified as POLE EDM after analysis. The authors believe that the p53 alteration was a passenger event that occurred after the first alteration. The number of MMRd-POLE EDM-p53 abn patients was too small to compare clinical data. Until more studies have been performed the authors suggest classifying these tumors as POLE EDM or MMRd if a pathogenic or non-pathogenic mutation is found respectively [38].

We have summarized the data for adjuvant treatment received and stage (FIGO) by molecular subgroup (Table 4). Léon-Castillo et al. [48] was excluded from the adjuvant treatment analysis since all patients had received either radiotherapy or chemoradiotherapy in the PORTEC-3 study. Adjuvant treatment data for Bosse et al. [34] was not available. Tumor stage was not available for Stello et al. [25]. Tumors attributed to the POLE EDM subgroup presented at a lower stage as reported earlier [35]. The stage at initial diagnosis seems to resemble the outcome of the molecular subgroups. Arguably the excellent and poor prognosis of the POLE EDM and p53 abn subgroup respectively is attributed to this fact. Interestingly, in the studies summarized POLE EDM patients have received the second-highest rate of adjuvant treatment (51.2%) after p53 abn (58.8%), which is likely attributed to the fact that POLE EDM tumors usually present as high-grade endometrioid endometrial cancer (EEC) and are classified as high-risk in the classical histo-pathological risk groups.

## 5. Ongoing Trial and Conclusions

The PORTEC 4a trial is the first active trial comparing the molecular classifier against conventional histopathological risk groups after standard surgical procedures. Patients with high-intermediate and high-risk up to Stage II according to the 2014 EMSO risk group are included [52]. After a laparoscopic or abdominal hysterectomy and bilateral salpingo-oophorectomy (with or without pelvic lymphadenectomy) patients are randomized in a 2:1 ratio for experimental and standard treatment respectively. The targeted study size is 500 patients. The experimental arm includes observation only for POLE mutated patients and p53 wt patients with no *CTNNB1* mutation which are expected to have a favorable outcome. MMRd and p53 wt patients with *CTNNB1* mutation are planned to receive vaginal brachytherapy and are grouped as intermediate. The group of p53 abn patients and any tumor that shows substantial LVSI or L1CAM expression > 10% is considered unfavorable and will receive external beam radiotherapy. In conclusion, even if PORTEC 4a does not strictly rely on the ProMisE algorithm, this trial combines ProMisE with several well-studied markers and attempts to determine its predictive value in comparison to the conventional histopathological risk classifier.

Since the discovery of the four EC clusters by TCGA in 2013, the development of a molecular classifier has made rapid progress and the results presented look promising. A limitation that must be noted is that the majority of the presented studies were published by only two groups (Vancouver and transPORTEC). From a purist point of view, until today no single randomized trial used the molecular classifier in order to stratify patients for adjuvant treatment modalities. Furthermore, well-established decisions on adjuvant treatment rely on several randomized clinical trials using conventional histopathological risk groups. On the other hand, the conventional risk classifier exhibits low inter-observer reliability, low reproducibility, and a low capacity to distinguish patients with high-grade EC from patients with favorable forms of EC. Up to 30% of cases with high-grade EC are misdiagnosed leading to impaired treatment decisions [8,9]. On the other side, the clinical data of over 3000 EC patients are available for interpretation. With only 3% of patients showing signs of multiple classifiers, ProMisE offers a high-level of unambiguity. Furthermore, the immunohistochemistry and mutation analysis does not rely on subjective interpretation resulting in a higher level of reproducibility. In the meantime, the molecular classification of EC was made more readily available in terms of cost and time. However, it still requires a much higher effort in its implementation compared to conventional histopathological diagnostics.

The 2016 ESMO-ESGO-ESTRO consensus conference has already mentioned the TCGA molecular classification for EC. In late 2020 the ESGO-ESTRO-ESP guidelines were updated, sharing the conclusions of this review, and encouraged the determination of the molecular subtype, and classification of all EC patients (Table 1). For patients with early POLE EDM EC, adjuvant treatment de-escalation may be feasible and is recommended to treat as low-risk, while p53 abn patients except stage IA are considered at high-risk and receive extended adjuvant treatment. For the MMRd and NSMP groups, the determination of other features such as LVSI and grading is employed to decide the individual patients’ risk and need for treatment. The incorporation of ProMisE in prospectively randomized clinical trials with the aim to determine the adjuvant treatment to TCGA is urgently warranted to overcome the shortcomings of conventional histopathological risk stratification. This could pave the way for TCGA molecular classification into the daily routine.

## Figures and Tables

**Figure 1 cancers-13-01478-f001:**
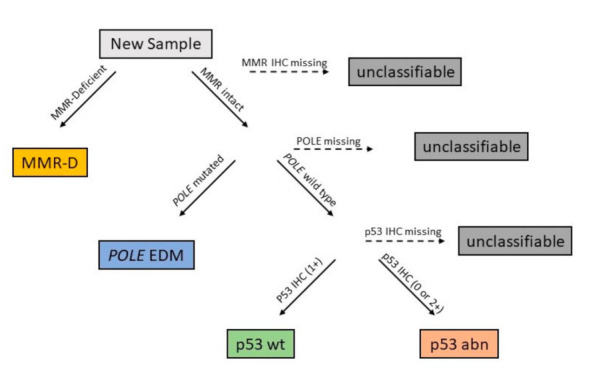
The Proactive Molecular Risk Classifier for Endometrial Cancer (ProMisE) Algorithm to assess a new endometrial cancer sample. First, mismatch-repair (MMR)-deficiency is evaluated with immunohistochemistry (IHC) against MSH 6 and PMS2 proteins. Second, the Polymerase Epsilon (*POLE)* exonuclease domain is tested by sequencing exons 9–14. Lastly, IHC for p53 is performed to determine patients with normal expression (IHC score 1+) versus complete loss/null (IHC score 0) or accumulation (IHC score 2+). Reproduced with permission from [31].

**Table 1 cancers-13-01478-t001:** 2020 ESGO/ESTRO/ESP prognostic risk group to guide adjuvant therapy use [11].

Risk Group	Molecular Classification Unknown	Molecular Classification Known
Low	Stage IA endometrioid, grade 1–2, LVSI negative or focal	Stage I–II POLE EDM endometrial carcinoma, no residual disease Stage IA MMRd/p53 wt endometrioid carcinoma + low grade + LVSI negative or focal
Intermediate	Stage IB endometrioid, grade 1–2, LVSI negative or focal Stage IA endometrioid, grade 3, LVSI negative or focal Stage IA non-endometrioid (serous, clear cell, undifferentiated carcinoma, carcinosarcoma, mixed) without myometrial invasion	Stage IB MMRd/p53 wt endometrioid carcinoma + low-grade + LVSI negative or focal Stage IA MMRd/p53 wt endometrioid carcinoma + high-grade + LVSI negative or focal Stage IA p53 abn and/or non-endometrioid without myometrial invasion
High-intermediate	Stage I endometrioid, substantial LSVI, regardless of grade and depth of invasion Stage IB endometrioid, grade 3, regardless of LVSI status Stage II	Stage I MMRd/p53 wt endometrioid carcinoma + substantial LVSI regardless of grade and depth of invasion Stage IB MMRd/p53 wt endometrioid carcinoma high-grade regardless of LVSI status Stage II MMRd/p53 wt endometrioid carcinoma
High	Stage III–IVA with no residual disease Stage I–IVA non-endometrioid (serous, clear cell, undifferentiated carcinoma, carcinosarcoma, mixed) with myometrial invasion, and with no residual disease	Stage III–IVA MMRd/p53 wt endometrioid carcinoma with no residual disease Stage I–IVA p53abn endometrial carcinoma with myometrial invasion, with no residual disease Stage I–IVA p53 wt/MMRd serous, undifferentiated carcinoma, carcinosarcoma with myometrial invasion, with no residual disease
Advanced	Stage III–IVA with residual disease	Stage III–IVA with residual disease of any molecular type
Metastatic	Stage IVB	Stage IVB of any molecular type

European Society of Gynaecological Oncology (ESGO), European SocieTy for Radiotherapy and Oncology (ESTRO), European Society of Pathology (ESP), lymphovascular space invasion (LVSI).

**Table 2 cancers-13-01478-t002:** Features of the four molecular subtypes. Adapted from [35].

Subtype (Synonyms)	POLE-Mutant POLE EDM	MMRd MSI	p53 wt, MSS, CN Low NSMP	p53 Abn CN High
Mutational frequency	>100 mutations/Mb	100–10 mutations/Mb	<10 mutations/Mb	<10 mutations/Mb
Somatic copy-number alterations	Very low	Low	Low	High
Top five recurrent gene mutations (%)	POLE (100%) DMD (100%) CSMD1 (100%) FAT4 (100%) PTEN (94%)	PTEN (88%) PIK3CA (54%) PIK3R1 (42%) RPL22 (37%) ARID1A (37%)	PTEN (77%) PIK3CA (53%) CTNNB1 (52%) ARID1A (42%) PIK3R1 (33%)	TP53 (92%) PIK3CA (47%) FBXW7 (22%) PPP2R1A (22%) PTEN (10%)
Associated histological feature	Endometrioid Grade 3 Ambiguous morphology Broad front invasion TILs, peri-tumoral Lymphocytes Giant tumoral cells	Endometrioid Grade 3 LVSI substantial MELF-type invasion TILs, Crohn’s-like peri-tumoral reaction lower uterine segment involvement	Endometrioid Grade 1–2 Squamous differentiation ER/PR expression	Serous Grade 3 LVSI Destructive invasion High cytonuclear atypia Giant tumoral cells Hobnailing, Slit-like spaces
Associated clinical features	Lower BMI Early Stage (IA/IB)Early onset	Higher BMI Lynch Syndrome	Higher BMI	Lower BMI Advanced stageLate onset
Prognosis in early stage (I–II)	Excellent	Intermediate	Excellent/intermediate/poor	Poor
Diagnostic test	Sanger/NGS Tumor mutation burden	MMR-IHC (MLH1, MSH2, MSH6, PMS2) MSI assay Tumor mutation burden		p53-IHC NGS Somatic copy-number aberrations

Microsatellite instability (MSI), microsatellite stable (MSS), no specific molecular profile (NSMP), copy-number (CN), tumor-infiltrating lymphocytes (TILs), immunohistochemistry (IHC), next-generation sequencing (NGS).

**Table 3 cancers-13-01478-t003:** Published trials using ProMisE.

Author	Patient Cohort	Number of Patients	FIGO Stages					Subtypes				HR OS Multivariable				HR RFS Multivariable			
			IA	IB	II	III	IV	POLE	MMRd	p53 wt	p53 abn	POLE	MMRd	p53 wt	p35 abn	POLE	MMRd	p53 wt	p53 abn
Stello 2015	Portec 3 criteria	116	36.2%		18.1%	35.3%	9.5%	12.1%	16.4%	37.9%	33.6%								
Talhouk 2015	“discovery”	143	71.3%		28.7%			8.4%	28.7%	44.1%	17.5%	0.28 (0.00–3.01)	0.90 (0.31–2.73)	1.00	4.28 (0.95–18.34)	0.15 (0.00–1.94)	0.32 (0.10–1.03)	1.00	1.64 (0.32–7.06)
Stello 2016	PORTEC 1 & 2	834	n/a		n/a			5.9%	26.3%	59.0%	8.9%	1.105 (0.394–3.101)	1.879 (1.307–2.700)	1.00	3.777 (2.364–6.037)				
Talhouk 2017	“confirmation”	319	69.3%		29.5%			9.4%	20.1%	27.0%	43.6%	1.01 (0.26–2.99)	1.90 (0.88-4.04)	1.00	2.61 (1.27–5.72)	0.19 (0.02–0.81)	0.64 (0.25–1.60)	1.00	1.75 (0.84–3.96)
Bosse 2018	Grade 3 EEC	381	44.9%	31.5%	30.2%	13.1%	2.9%	12.9%	36.2%	30.2%	20.7%	0.56 (0.27–1.15)	0.84 (0.57-1.25)	1.00	1.37(0.9–2.09)	0.23 (0.07–0.77)	0.61 (0.37–1.00)	1.00	1.92 (1.20–3.07)
Cosgrove 2018	NRG/GOG GOG210	982	74,5%		9.3%	14.4%	1,8%	4.0%	38.6%	48.9%	8.6%	0.19 (0.03–1.35)	1.04 (0.70–1.56)	1.00	1.61 (0.93–2.78)	0.26 (0.06–1.05)	1.08 (0.78–1.50)	1.00	1.56 (0.99–2.48)
Kommoss 2018	“validation”	452	61.1%	19.7%	5.8%	12.2%	1.3%	9.3%	28.1%	50.4%	12.2%	0.95 (0.30–2.36	1.41 (0.82–2.41)	1.00	2.29 (1.12–4.65)	0.15 (0.00–n/a)	1.54 (0.73–3.24)	1.00	3.40 (1.30–8.81)
León-Castillo 2020	PORTEC 3	423	13.2%	17.8%	25.6%	43.4%		12.4%	33.4%	31.5%	22.7%	0.118 (0.016–0.868)	1.00	0.547 (0.302–0.993)	2.298 (1.418–3.726)	0.079 (0.011–0.576)	1.00	0.976 (0.620–1.537)	2.517 (1.621–3.907)
Total		3650						7.9%	30.9%	46.5%	14.7%								

Endometrioid endometrial cancer (EEC), hazard ratio (HR), overall survival (OS), recurrence-free survival (RFS), not available (n/a)

**Table 4 cancers-13-01478-t004:** Analysis of the International Federation of Gynecology and Obstetrics (FIGO) stage and adjuvant treatment distribution in terms of the patients’ molecular subtype out of the published trials [25,30,31,33,34,46,48].

	Total	POLE EDM	MMRd	p53 wt	p53 abn	*p*-Value
*n*	%	*n*	%	*n*	%	*n*	%	*n*	%
Adjuvant treatment											
Any	1283	47.3	87	51.2	385	46.5	624	44.6	187	58.8	<0.001
None	1432	52.7	83	48.8	443	53.5	775	55.4	131	41.2	
Stage											
I	1838	68.7	187	84.6	581	65.7	831	72.3	241	56.9	<0.001
II–IV	838	31.3	34	15.4	302	34.3	319	27.7	181	43.1

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
