# Peer review of "The TCGA Molecular Classification of Endometrial Cancer and Its Possible Impact on Adjuvant Treatment Decisions"

_cancers, 2021, doi:10.3390/cancers13061478_

Round 1
Reviewer 1 Report
The manuscript submitted to Cancers entitled “The TCGA molecular classification and its possible impact on 2 adjuvant treatment decisions” is a well-written and comprehensive review, that is very actual and of high clinical relevance. Nonetheless I have several suggestions and requests regarding that manuscript:
- I would highly recommend including the term “endometrial cancer” in the title of the paper!
Abstract line 5: MMR deficient and not “MMR deficiency”
Introduction: one typo: Bokhman instead of “Bohman”.
Table 3 is by far overcharged; and should be cleared out! In its present form it is confusing.
In my opinion the section 4 of the paper is focusing too much on the prevalence reported in the various studies (numbers that are already given in Table 3) and other important aspects are lacking (such as, whether or to what percentage the reported POLE mutated cancers have been adjuvantly treated in the respective cohorts). Perhaps authors can obtain these data directly from the authors of the referenced studies? For PORTEC-3 it would be interesting to know how many patients were treated in the experimental and control arm.
Reviewer 2 Report
This is a well-written paper summarizing the papers that have been published on ProMisE classification in endometrial cancer and outcome. As mentioned, none of the studies were randomized with respect to the ProMisE classifcation. The summarized papers are actually mainly of two groups (Portec and Vancouver) which is a limitation that might need to be addressed. In addtion, there is no information on the stage in relation to the ProMisE which might have added new information. As demonstrated by Kommoss paper, about 50% of LNM are attributed to the abnormal p53 status, and hence might be an important explanation why this group has such a poor outcome. The Portec 1-2 studies included patients without information on nodal status, which might have resulted in an underestimation of occult stage III. So far nobody has actually shown the KM for tumor stage (I-II-III-IV) for each ProMisE group separately which still leaves this question unresolved whether properly staged, node negative (or sentinel node) with abnormal P53 have comparable outcome as stage IIIC with abnormal 53.
By adding these data, there would be new and additional information
Reviewer 3 Report
The study summarized the information reasonably well. But the authors have limited understanding about the essence of ProMiSe classifier. It is too early to apply the classifier into the adjuvant treatment decision. Here are several specific points for authors' consideration.
1) POLE mutation analysis represents a molecular test, which is currently not available in more than 99% of the pathology laboratory. This test represents the most important step in the ProMise algorithm. When this step can't be performed properly, the classifier is useless. Meanwhile, the test requires standardization in the national and international level. Some studies used whole genome sequencing analysis, while others used a few exons between exons 9 to 13. It is uncertain if all these tests have the same clinical value. Additionally, definition of POLE ultramutaion has not been completely settled.
2) It is well-recognized that Endometrial cancer TCGA (2013) molecular classification is much better than daily used histologic evaluation with the aid of IHC stains. However, the world is not ready to apply the TCGA classification to all endometrial cancer specimen. As a matter of fact, molecular analysis is not just using a piece of tumor tissue to run a molecular profile. Endometrial cancers are composed by heterogenous components. More importantly, within the copy number high group, the TCGA classification did not separate endometrioid carcinoma with p53 mutation and endometrial serous carcinoma with p53 mutation, which are known to have the prognostic differences. This is actually an important drawback of the original study in 2013 by Levine et al published in Nature.
Considering these facts, I think many studies are needed prior to the application of the ProMise classifier. That is to say, the ProMise classifier is only a promise, not ready to be applied in clinic practice.
Reviewer 4 Report
The authors present a review that summarizes the evolution of the proposal
and validation of the Proactive Molecular Risk Classifier for Endometrial Cancer
(ProMisE) as an algorithm for attributing patients with endometrial cancer
to different risk groups.
The review is well organized and presented and the language is fluid and
easy to understand.
I have only one point to bring to the attention of the authors by suggesting an
update of the conclusions of their review.
In December 2020, the latest ESGO / ESTRO / ESP guidelines for the management
of patients with endometrial cancer were published Online First on Int J Gynecol
Cancer. In these guidelines the indications for adjuvant treatment have already
been adapted to the case in which the molecular classification is known for the
patient. Another small correction is needed on page 1 line 27, where "Bokhman" is written
"Bohman".
Reviewer 5 Report
In this review the authors summarize the development of the TGCA molecular classification for endometrial cancer and its possible impact on adjuvant treatment decisions. The article is well written and referenced.
Suggestions:
Please reference and elaborate on the new histopathological WHO classification of EC
Please illustrate the WHO algorithm for molecular classification (maybe a Figure 1B) and discuss differences/ implications vs PROMISE algorithm
Please check English grammar / spelling
Round 2
Reviewer 2 Report
I think the authors have properly addressed the issues and limitations of the performed studies on Proactive Molecular Risk Classifier for Endometrial Cancer (ProMisE). The added Table 4 is very illustrative to the data and the current gaps in knowledge